# Novel Mutation in the Feline *NPC2* Gene in Cats with Niemann–Pick Disease

**DOI:** 10.3390/ani13111744

**Published:** 2023-05-24

**Authors:** Tofazzal Md Rakib, Md Shafiqul Islam, Mohammad Mejbah Uddin, Mohammad Mahbubur Rahman, Akira Yabuki, Tetsushi Yamagami, Motoji Morozumi, Kazuyuki Uchida, Shinichiro Maki, Abdullah Al Faruq, Osamu Yamato

**Affiliations:** 1Laboratory of Clinical Pathology, Joint Faculty of Veterinary Medicine, Kagoshima University, Korimoto, Kagoshima 890-0065, Japan; rakibtofazzal367@gmail.com (T.M.R.); si.mamun@ymail.com (M.S.I.); mmu_cvasu@yahoo.com (M.M.U.); mahbub57991@yahoo.com (M.M.R.); yabu@vet.kagoshima-u.ac.jp (A.Y.); k6993382@kadai.jp (S.M.); faruqabdullahal103@gmail.com (A.A.F.); 2Faculty of Veterinary Medicine, Chattogram Veterinary and Animal Sciences University, Khulshi, Chattogram 4225, Bangladesh; 3Japan Small Animal Medical Center, Saitama, Tokorozawa 359-0023, Japan; t_yamagami@mac.com; 4Togasaki Animal Hospital, Saitama, Misato 341-0044, Japan; morozumi@togasaki-ah.com; 5Laboratory of Veterinary Pathology, Graduate School of Agricultural and Life Sciences, University of Tokyo, Bunkyō, Tokyo 113-8657, Japan; auchidak@g.ecc.u-tokyo.ac.jp

**Keywords:** Niemann–Pick disease type C, feline *NPC2* gene, lysosomal storage disease, Siamese cat, Japanese domestic cat

## Abstract

**Simple Summary:**

Niemann–Pick disease (NP) type C is an autosomal recessive metabolic disorder caused by mutations in the *NPC1* or *NPC2* gene. It is characterized by cholesterol accumulation in the lysosomes and late endosomes. Herein, we studied the molecular basis of Siamese and Japanese domestic (JD) cats that were previously diagnosed with NP. Sanger sequencing was performed on all exons of the two genes using genomic DNA extracted from the paraffin-embedded tissues of these cats. As a result, a missense mutation (*NPC2*:c.376G>A, p.V126M) was identified as a candidate pathogenic mutation in both types of cats. Several pathogenicity and stability predictors showed that this mutation was deleterious and severely decreased NPC2 protein stability. The Siamese cat was homozygous and the JD cat was heterozygous for this mutation. No other exonic *NPC2* mutations were detected in the JD cat; however, a homozygous splice variant (c.364-4C>T) was identified, which is not known to be associated with this disease.

**Abstract:**

Niemann–Pick disease (NP) type C is an autosomal, recessive, and inherited neurovisceral genetic disorder characterized by the accumulation of unesterified cholesterol and glycolipids in cellular lysosomes and late endosomes, with a wide spectrum of clinical phenotypes. This study aimed to determine the molecular genetic alterations in two cases of felines with NP in Japan, a Siamese cat in 1989 and a Japanese domestic (JD) cat in 1998. Sanger sequencing was performed on 25 exons of the feline *NPC1* gene and 4 exons of the feline *NPC2* gene, using genomic DNA extracted from paraffin-embedded tissue specimens. The sequenced exons were compared with reference sequences retrieved from the GenBank database. The identified mutations and alterations were then analyzed using different prediction algorithms. No pathogenic mutations were found in feline *NPC1*; however, c.376G>A (p.V126M) was identified as a pathogenic mutation in the *NPC2* gene. The Siamese cat was found to be homozygous for this mutation. The JD cat was heterozygous for the same mutation, but no other exonic *NPC2* mutation was found. Furthermore, the JD cat had a homozygous splice variant (c.364-4C>T) in the *NPC2* gene, which is not known to be associated with this disease. The *NPC2*:c.376G>A (p.V126M) mutation is the second reported pathogenic mutation in the feline *NPC2* gene that may be present in the Japanese cat population.

## 1. Introduction

Niemann–Pick disease (NP) encompasses a group of autosomal recessive lysosomal storage disorders characterized by the accumulation of diverse lipid species in lysosomes [1]. NP types A (infantile neurovisceral type, MIM #257200) and B (chronic visceral type, #607616) are caused by deficits in the activity of acid sphingomyelinase, an enzyme that regulates lysosomal sphingomyelin homeostasis, which is encoded by the *SMPD1* gene. NP type C (NPC) is caused by mutations in the *NPC1* (#257220) and *NPC2* genes (#607625), resulting in functional defects in lysosomal proteins NPC1 and NPC2, which are involved in cholesterol efflux from lysosomes.

NPC is an inherited neurovisceral genetic disorder characterized by the accumulation of unesterified cholesterol and glycolipids in cellular lysosomes and late endosomes, with multiple clinical phenotypes [2]. NPC is divided into two types of diseases, C1 and C2, based on the mutations in *NPC1* or *NPC2*. Both NPC proteins, NPC1 and NPC2, are required for sterol homeostasis and are thought to be responsible for sterol integration into the lysosomal membrane before its redistribution to other cellular membranes [3]. A soluble intralysosomal protein, NPC2, can transfer sterols to the luminal *N*-terminal domain of the lysosomal membrane protein NPC1 with reversed polarity by binding to sterols with distinct polarity [4]. The intracellular trafficking of lipids such as cholesterol, lipophilic molecules, amines, and mycolic acids is mediated by NPC1 and NPC2 proteins [5]. Because of the functions of these proteins, when one or both are lacking or unstable, the lysosome/late endosome compartment begins to accumulate endocytosed unesterified cholesterol, gangliosides, and other lipids [3,6,7]. Therefore, NPC disorders stem from mutations in *NPC1* and *NPC2*, which affect quality control mechanisms involving the lysosome and endoplasmic reticulum [7].

In humans diagnosed with NPC, 133 *NPC1* and 11 *NPC2* pathogenic variants have been described [8,9]. Because of the functions of these genes in lysosomes and late endosomes, the predominant biochemical characteristic of NPC is the lysosomal accumulation of unconjugated cholesterol and glycosphingolipids in the brain, liver, and other visceral organs [1,7,8]. Mitochondrial cholesterol buildup and dysfunction are also observed, which results in the impaired transport of cytosolic reduced glutathione (GSH) into the mitochondria. Moreover, a decreased level of mitochondrial GSH causes oxidative stress in the cells, and neurons and hepatocytes may die as a consequence of the loss of antioxidants, such as GSH [7]. Therefore, neurovisceral cholesterol storage with progressive neuronal cell loss, axonal spheroids, and ectopic neurites, are key microscopic findings in NPC. Clinical manifestations include progressive neurological symptoms such as dysmetria, ataxia, and generalized tremors, as well as significantly higher levels of unconjugated cholesterol in neurovisceral organs [2,9]. The phenotypes of human NPC vary greatly, and manifest from fetal life to adulthood. Most patients experience a progressive and deadly neurological condition. However, an exception of this is found in those who have the perinatal form of the disease, which commonly results in death and appears at birth or during the first month of life with significant visceral involvement, including fetal hydrops, ascites, neonatal cholestasis, liver failure, and/or pulmonary disease [6,10].

NPC has also been described in animals, including a Boxer dog (Type C1, OMIA 000725-9615) [11], cats (Type C1, OMIA 000725-9685; Type C2, OMIA-002065-9685) [6,12,13,14], and Angus cattle (Type C1, OMIA 000725-9913) [15,16]. In cats, two pathogenic mutations, c.2864G>C (p.C955S) [14] and c.1322A>C (p.H441P) [13], in the feline *NPC1* gene, and one pathogenic mutation, c.82+5G>A (p.G28_S29ins35) [6], in the feline *NPC2* gene, have been reported. Additionally, one pathogenic mutation, c.2969C>G (p.P990R), has been identified in bovine *NPC1* [15]. No such pathogenic mutations have been identified in other animals, including dogs.

To date, very few cases of feline NPC have been reported [6,13,14]. In Japan, a Siamese cat and a Japanese domestic (JD) cat suspected of having NPC were described in 1989 [16,17,18] and 1998 [19], respectively. Recently, another Siamese cat was diagnosed with NP and proven to be affected by NP type A because the cat was homozygous for a novel nonsense mutation (*SMPD1*:c.1017G>A (p.W339*)) [20]. The first two cases were diagnosed as NP based on clinical manifestations and histopathological, ultrastructural, and tissue biochemical analyses, which were very similar to those of human and feline NPC cases [16,17,18]. Fortunately, some of the paraffin-embedded tissue specimens were stored for a long time, and could be analyzed molecularly. This study aimed to identify feline-specific pathogenic mutation(s) in cats with NPC.

## 2. Materials and Methods

This study was performed in accordance with the Guidelines Regulating Animal Use and Ethics of Kagoshima University (No. VM15041; approval date: 29 September 2015). Informed oral consent was also obtained from the owners of the cats with NPC [16,19].

### 2.1. Specimen

For the DNA sequencing analysis, specimens were obtained from the paraffin-embedded liver and brain samples of two cats diagnosed with NP based on clinical, pathological, ultrastructural, and biochemical findings [16,17,18,19]. Genomic DNA was extracted from the specimens using a QIAamp DNA Mini Kit (Qiagen, Hilden, Germany) and automated extraction equipment (magLEAD 6gC; Precision System Science, Co., Ltd., Matsudo, Japan) according to the manufacturer’s recommendations.

### 2.2. Mutation Analysis

The coding exons and splice junctions of the feline *NPC1* and *NPC2* genes were amplified and assessed using polymerase chain reactions (PCR), and Sanger sequencing was performed using specific primer pairs (Appendix A). The primer pairs were designed based on the reference sequences XM_019814307.3 (*NPC1*) and XM_003987833.6 (*NPC2*); the need arose because the DNA from the paraffin-embedded samples was modestly fragmented. PCR was performed in a 20 µL reaction mixture containing 10 µL 2X PCR master mix (GoTaq Hot Start Green Master Mix, Promega Corp., Madison, WI, USA). The PCR products were purified before sequencing using a QIAquick Gel Extraction Kit (Qiagen) according to the manufacturer’s instructions. Sanger sequencing was performed by Kazusa Genome Technologies Ltd. in Kisarazu, Japan. The obtained sequencing data were compared with the above-mentioned reference sequences to identify candidate pathogenic mutations. Multiple runs were performed in both directions for each cat to generate a consensus sequence. Sequence data were obtained from the two affected cats and four clinically healthy JD cats. Before analyzing *NPC1* and *NPC2* in the two cats, we confirmed the absence of the *SMPD1*:c.1017G>A (p.W339*) mutation, which causes NP type A [20].

### 2.3. Pathogenicity and Stability Prediction

Pathogenicity and the changes in protein stability upon mutations were analyzed using the PredictSNP and iStable servers, respectively. The PredictSNP server has an embedded algorithm for PredictSNP, MAPP, PhD-SNP, Polyphen-1, Polyphen-2, SIFT, and SNAP to classify variants as either “deleterious” or “neutral” [21]. Similarly, iStable has an embedded algorithm for iStable, i-Mutant, and MUpro to classify the variants as either “increasing” or “decreasing” the stability of the proteins [22,23,24]. I-Mutant2.0 also gives the delta delta G (ddG) value, which is an indicator of the difference in free energy caused by the mutations. The amino acid sequences in the FASTA format and mutations were used as inputs for prediction. To compare and validate the pathogenicity of the identified candidate missense mutations, we also analyzed several pathogenic missense mutations that have already been reported in humans (*NPC1*:c.2974G>T, c.3019C>G, and c.3182T>C; *NPC2*:c.115G>A and c.358C>T) [8], cats (*NPC1*:c.1322A>C and c.2864G>C) [13,14], and cattle (*NPC1*:c.2969C>G) [15].

### 2.4. Preserving Property

The preserving property of the amino acid sequence of the feline *NPC2* gene was analyzed and compared with that of several other animals using the Protein Basic Local Alignment Search Tool (Protein BLAST) provided by the National Center for Biotechnology Information (NCBI) site.

### 2.5. Pathogenicity Prediction of the Splice Variant

The identified splice variant was analyzed for pathogenicity using IntSplice ver. 2.0 [25]. IntSplice predicts a splicing consequence of a single nucleotide variation at intronic positions −50 to −3 close to the 3′-end of an intron of the human genome. However, we were unable to perform predictions using the feline genome because of the lack of online prediction algorithms. The position of the same nucleotide in the human genome was determined and predicted using a pathogenicity server, and a pathogenic probability score (>0.5) was considered pathogenic [25].

### 2.6. Preliminary Survey

A preliminary survey was conducted on variants identified in this study using the 99 Lives Consortium dataset coordinated by the Lyons Feline Genetics Laboratory at the University of Missouri.

## 3. Results

### 3.1. Mutation Detection

Sanger sequencing was performed on all the exons and splice junctions of the feline *NPC1* and *NPC2* genes in a Siamese cat and a JD cat diagnosed with NP. The sequenced exons and splice junctions were compared with the reference sequences XM_019814307.3 (*NPC1*) and XM_003987833.6 (*NPC2*). As a result, the sequencing and comparison in the feline *NPC1* gene revealed three homozygous silent mutations (c.54G>T (p.A18A), c.1815T>C (p.T605T), c.2214C>A (p.S605S)), one heterozygous silent mutation (c.2043C>T (p.P681P)), and one homozygous missense mutation (c.3224G>A (p.R1075H)) in the Siamese cat; nevertheless, no change was found in the JD cat. In the feline *NPC2* gene, one homozygous silent mutation (c.198A>C (p.S66S)) and one homozygous missense mutation (c.376G>A (p.V126M)) were identified in the Siamese cat (Figure 1). The JD cat was heterozygous for the same mutation (c.376G>A (p.V126M)) (Figure 1) and homozygous for a splice junction variant (c.364-4C>T (g.121865226C>T)) (Figure 2).

### 3.2. Pathogenicity and Stability Prediction

The two missense mutations (*NPC1*:c.3224G>A (p.R1075H) and *NPC2*:c.376G>A (p.V126M)) identified in the two affected cats were further analyzed using the SIFT and PredictSNP servers, which included seven prediction algorithms (Table 1). In addition, we included five pathogenic mutations (three *NPC1* and two *NPC2* mutations) reported in humans, and two feline mutations and one bovine *NPC1* mutation reported in animals as a reference. The *NPC1*:c.3224G>A (p.R1075H) mutation was predicted to be neutral by six predictors, except for SIFT which had a marginal score (0.05). In contrast, the *NPC2*:c.376G>A (p.V126M) mutation was predicted to be pathogenic by all pathogenicity predictors, with a very low SIFT score (0.00). Other mutations already reported in humans and animals showed a deleterious effect on the catalytic site of the NPC1 or NPC2 protein in the analysis of all seven or at least five pathogenicity predictors.

Based on the results of all three stability predictors, analysis of the predicted stability using the iStable server showed decreased stability of the NPC2 protein produced by the *NPC2*:c.376G>A (p.V126M) mutation (Table 2). Additionally, two of three stability predictors showed decreased stability of the NPC1 protein due to the *NPC1*:c.3224G>A (p.R1075H) mutation. Among the other mutations reported in humans and animals, three human mutations (*NPC1*:p.1007A and p.I1061T and *NPC2*:p.P120S) and a bovine *NPC1*:c.2969C>G (p.P990R) mutation showed decreased stability in all three predictors. However, two human mutations (*NPC1*:p.G992W and *NPC2*:pV39M) were predicted to decrease the stability by one and two predictors, respectively. Among the reported feline *NPC1* mutations, the c.2846G>C (p.C955S) mutation was predicted to decrease stability based on two predictors. However, the feline *NPC1*:c.1322A>C (p.H441P) mutation was not predicted to decrease the stability by any predictor, but two predictors predicted an increase in the stability.

In association with the feline *NPC2*:c.376G>A (p.V126M) mutation, the preserving property of an amino acid residue of p.V126 was surveyed in a variety of animals using Protein BLAST (Figure 3). As a result, valine at position 126 was found to be highly preserved in many mammals including dogs, cattle, sheep, horses, dolphins, humans, macaques, mice, and tigers.

### 3.3. Splice Variant Analysis

A homozygous variant in the splice region (*NPC2*:c.364-4C>T), identified in a JD cat with NP, was analyzed using IntSplice ver. 2.0. This variant was predicted to be normal with a low pathogenic probability score (0.106938).

### 3.4. Preliminary Survey

The preliminary survey was conducted on two missense variants (*NPC1*:c.3224G>A (p.R1075H) and *NPC2*:c.376G>A (p.V126M)) and a splice variant (*NPC2*:c.364-4C>T), using the 99 Lives Consortium dataset that currently contains whole-genome sequence data from 362 domestic cats and whole-exome sequence data from 62 cats. The survey result showed that these three variants were not present in this dataset.

## 4. Discussion

Clinical manifestations, serum biochemistry, and imaging tests such as magnetic resonance imaging are common tools for the premortem diagnosis of lysosomal storage diseases, except when the molecular causes are known [12]. These clinical diagnostic tools are not enough to differentially diagnose metabolic diseases from other related diseases with similar clinical manifestations. Necropsy findings and further pathological and laboratory investigations allow us to approach the diagnosis, but it is still difficult to diagnose genetic diseases without any molecular diagnostic tools, such as PCR, especially in the veterinary field. This is because of insufficient information about the molecular bases of genetic diseases in animals. Thus, once a causative mutation is identified in an animal, breed, or family, simple procedures can be used for rapid genetic diagnosis and genotype screening, leading to effective disease control and the eradication of the disease [26,27]. For this purpose, stored paraffin-embedded specimens of animals suspected of having genetic diseases can be utilized if a tentative diagnosis has already been established. We recently used DNA extracted from a paraffin-embedded specimen of a cat with a genetic disease, and successfully identified the first pathogenic mutation in feline Pompe disease, which is a glycogen storage disease type II [28].

Two cats suspected to have NPC were described in Japan in 1989 [16,17,18] and 1998 [19]. The first case involved a 6-month-old female Siamese cat that showed neurological signs such as an ataxic gait, head tremors, loss of equilibrium, and hepatomegaly [16]. This cat died at 11 months of age due to the progression of neurological dysfunction, and was diagnosed with neurovisceral sphingomyelinosis postmortem, with an accumulation of other types of lipids based on lipid analysis, histopathology, electron microscopy, and lectin histochemistry [16,17,18]. The second case was a 3-month-old male JD cat with progressive neurological signs such as ataxia, generalized tremors, and impaired vision. The JD cat showed increased liver enzyme activity in serum biochemistry and had brain atrophy, as suggested by computed tomography. The cat died at 5 months of age due to the progression of neurological dysfunction. Postmortem, the disease proved to be neurovisceral NP with an accumulation of sphingomyelin and cholesterol based on lipid analysis, histopathology, and electron microscopy [19]. The clinical, biochemical, and histopathological characteristics of these two cats with NP were similar to those of NPC rather than those of NP type A. Consequently, we analyzed the feline *NPC1* and *NPC2* genes after confirming the absence of the *SMPD1*:c.1017G>A (p.W339*) mutation that was recently reported in a Siamese cat with NP type A [20].

Sanger sequencing of the feline *NPC1* and *NPC2* genes of the two cats with NP, and their comparison with the reference sequences, were performed to identify candidate pathogenic mutations in feline NPC. As the candidates of potential pathogenic mutations, two missense mutations (*NPC1*:c.3224G>A (p.R1075H) and *NPC2*:c.376G>A (p.V126M)) and one splice variant (*NPC2*:c.364-4C>T (g.121865226C>T)) were found in the Siamese and/or JD cats (Figure 1 and Figure 2). After the evaluation of the pathogenicity and stability prediction (Table 1 and Table 2) with the comparison of the reference data obtained from several known pathogenic mutations in humans and animals, the *NPC2*:c.376G>A (p.V126M) mutation was selected as the most likely candidate for pathogenic mutation in these cats with NPC. This is because this mutation was evaluated to be deleterious by all seven pathogenicity predictors, and was observed to decrease the stability of all three predictors. In contrast, the *NPC1*:c.3224G>A (p.R1075H) mutation was ruled out as a pathogenic mutation based on the evaluation data (mostly neutral) obtained from the pathogenicity predictors (Table 1), although two of the three stability predictors showed decreased stability (Table 2). Furthermore, the *NPC2*:c.364-4C>T (g.121865226C>T) splice variant identified in the JD cat was suspected to be benign based on the splice variant analysis data; however, further studies are required to determine the relationship between this variant and NPC.

The preliminary survey using the 99 Lives Consortium dataset indicated that *NPC1*:c.3224G>A, *NPC2*:c.376G>A, and *NPC2*:c.364-4C>T were not present in the dataset, suggesting that the three variants may be rare in the domestic cat population and could be responsible for feline NPC. However, the 99 Lives Consortium dataset does not include data from many Asian cats; therefore, a large-scale survey of the Asian cat population would help demonstrate the exact frequencies of these variants.

The Siamese cat was homozygous for the *NPC2*:c.376G>A (p.V126M) mutation, whereas the JD cat was heterozygous for the same mutation. Therefore, the JD cat may be a compound heterozygote with two different mutations. A counterpart pathogenic mutation of *NPC2* may be located in the intronic regions that could not be analyzed in this study. When comparing the clinical manifestations of the two NPC-affected cats, the disease onset in the JD cat was earlier (at three months old) than that in the Siamese cat (at six months old) [16,19]. Furthermore, the JD cat died at an earlier age (five months old) than the Siamese cat (11 months). This suggests that the disease progression was faster in the JD cat than in the Siamese cat. Therefore, another mutation in the JD cat may have been as deleterious as a null mutation, such as a splice defect. In the first report of two littermate cats with NPC2 disease caused by a splice defect (*NPC2*:c.82+5G>A (p.G28_S29ins35)), the disease onset was at approximately three months old, and one cat died at 10 months of age, but the other littermate was alive for more than one year until euthanasia [6]. The clinical severity of these littermates with NPC2 disease was similar to that of our JD cat, rather, but not to that of the Siamese cat in this study. This also suggests that the JD cat may have had a null mutation as a counterpart mutation of *NPC2*:c.376G>A (p.V126M).

Treatment efforts for human NPC are currently focused on slowing disease progression [29]. The only registered drug that is promising for the treatment of patients with NPC is *N*-butyldeoxynojirimycin (miglustat), which was initially approved for the treatment of Gaucher disease, a hereditary deficiency of lysosomal glucocerebrosidase. Miglustat is an iminosugar that inhibits glucosylceramide synthase, which is required in the early stages of glycosphingolipid synthesis. The oral administration of miglustat (1200 mg/kg/day) in mice with NPC (BALBc/NPC^nih^) reduced the accumulation of gangliosides in the brain, slowed their neurological progression, and increased their lifespan by approximately 33% [30]. Subsequent studies on cats with NPC caused by *NPC1*:c.2864G>C (p.C955S) showed that the oral administration of miglustat (50 mg/kg/day) was effective in reducing the accumulation of gangliosides in the brain, delaying their onset of neurological symptoms, and increasing their lifespan by approximately 74% [30]. Other effective treatments are still under development, and combination therapies using several compounds and other strategies, such as gene therapy, are increasingly being worked on [29].

A colony of cats with NPC caused by *NPC1*:c.2864G>C (p.C955S) is kept at Colorado State University [29,30]. As described above, the cats are being utilized as a large animal model for NPC studies, including the development of new therapeutic methods [30]. However, to the best of our knowledge, a cat colony with NPC2-derived disease has not yet been established. Notably, there are differences between NPC1- and NPC2-derived diseases that require both animal models for an accurate study. We suppose that cats carrying the *NPC2*:c.376G>A (p.V126M) mutation are present in the cat population in Japan, which could be instrumental in future therapeutic research as a large animal model for human NPC2 disease.

## 5. Conclusions

This study identified a pathogenic mutation (*NPC2*:c.376G>A (p.V126M)) in the feline *NPC2* gene of a Siamese cat and a JD cat with NP. The pathogenicity of this mutation was confirmed based on the data obtained from a variety of predictors by comparing data from several known pathogenic mutations. The Siamese and JD cats were homozygous and heterozygous for this mutation, respectively. No other exonic *NPC2* mutation or deleterious splice variant was found in the JD cat, suggesting that this cat was a compound heterozygote of the identified *NPC2*:c.376G>A mutation and another pathogenic mutation that may be located in intronic regions. In conclusion, the c.376G>A (p.V126M) mutation is the second reported pathogenic mutation in the feline *NPC2* gene that causes NPC, and it may be present in the Japanese cat population.

## Figures and Tables

**Figure 1 animals-13-01744-f001:**
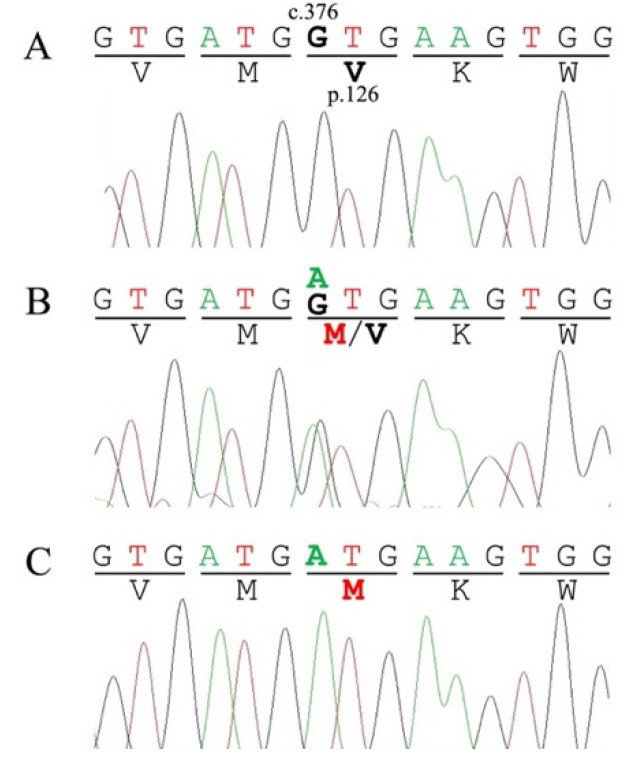
Partial genomic sequence electropherograms of exon 4 in the feline *NPC2* gene from a clinically healthy control cat (**A**), a Japanese domestic cat (**B**) with Niemann–Pick disease type C (NPC), and a Siamese cat with NPC (**C**). The control cat is a homozygous wild-type (c.376G/G). The Japanese domestic and Siamese cats are heterozygous (c.376G/A) and homozygous (c.376A/A) for the c.376G>A (p.V126M) mutation, respectively.

**Figure 2 animals-13-01744-f002:**
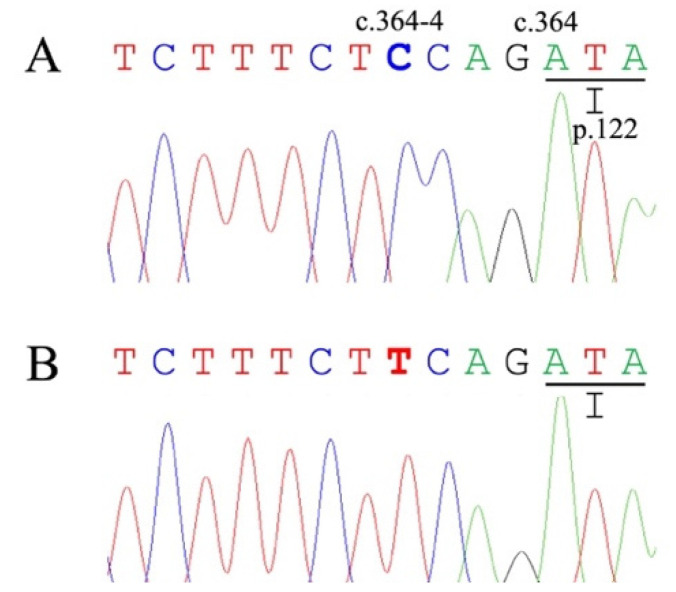
Partial genomic sequence electropherograms of the 5′-end of exon 4 in the feline *NPC2* gene from a clinically healthy control cat (**A**) and a Japanese domestic cat (**B**) with Niemann–Pick disease type C (NPC). The control cat is a homozygous wild-type (c.364-4C/C). The Japanese domestic cat is homozygous (c.364-4T/T) for the c.364-4C>T variant.

**Figure 3 animals-13-01744-f003:**
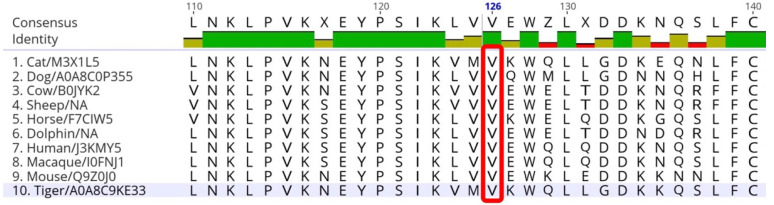
Multiple amino acid sequence alignment at position p.126V in the *NPC2* gene of different mammals. Valine at position 126 is highly preserved in these mammals.

**Table 1 animals-13-01744-t001:** Analysis of missense mutations in the catalytic site of the NPC1 and NPC2 proteins using the SIFT and PredictSNP servers.

Gene	Species	Mutation		SIFT		Predict SNP	MAPP	PHD-SNP	Polyphen-1	Polyphen-2	SNAP
		Nucleotide	Amino Acid	Score	Interpretation						
*NPC1*	Cat *	c.3224G>A	p.R1075H	0.05	N/D	N	N	N	N	N	N
	Cat	c.1322A>C	p.H441P	0.25	N	D	D	D	D	D	D
	Cat	c.2864G>C	p.C955S	0.00	D	D	D	D	D	D	N
	Cattle	c.2969C>G	p.P990R	0.00	D	D	D	D	D	D	D
	Human	c.2974G>T	p.G992W	0.03	D	D	N	D	D	D	N
	Human	c.3019C>G	p.P1007A	0.00	D	D	D	D	N	D	D
	Human	c.3182T>C	p.I1061T	0.02	D	D	D	D	D	D	N
*NPC2*	Cat **	c.376G>A	p.V126M	0.00	D	D	D	D	D	D	D
	Human	c.115G>A	p.V39M	0.01	D	D	D	D	D	D	D
	Human	c.358C>T	p.P120S	0.00	D	D	D	D	D	N	D

* *NPC1*:c.3224G>A (p.R1075H) mutation was identified in a Siamese cat with Niemann–Pick disease type C (NPC) in this study. ** *NPC2*:c.376G>A (p.V126M) mutation was identified in Siamese and Japanese domestic cats with NPC in this study. The other mutations have also been reported to be pathogenic for NPC in cats, cattle, and humans. D: deleterious, N: neutral, and N/D: marginal. SIFT scores (<0.05) were predicted to be deleterious and affect protein function.

**Table 2 animals-13-01744-t002:** Prediction of stability of NPC1 and NPC2 proteins for selected missense mutations by protein stability predictors.

Gene	Species	Mutation		i-Mutant2.0		MUpro		iStable	
		Nucleotide	Amino Acid	Result	ddG	Result	Conf. Score	Result	Conf. Score
*NPC1*	Cat *	c.3224G>A	p.R1075H	Decrease	−1.83	Null	Null	Decrease	0.78063
	Cat	c.1322A>C	p.H441P	Increase	0.35	Null	Null	Increase	0.631442
	Cat	c.2864G>C	p.C955S	Decrease	−0.59	Null	Null	Decrease	0.556555
	Cattle	c.2969C>G	p.P990R	Decrease	−0.47	Decrease	−0.093720996	Decrease	0.848305
	Human	c.2974G>T	p.G992W	Increase	0.10	Decrease	−0.13709342	Increase	0.573621
	Human	c.3019C>G	p.P1007A	Decrease	−1.16	Decrease	−1	Decrease	0.681309
	Human	c.3182T>C	p.I1061T	Decrease	−2.28	Decrease	−1	Decrease	0.826884
*NPC2*	Cat **	c.376G>A	p.V126M	Decrease	−1.03	Decrease	−0.96085544	Decrease	0.799741
	Human	c.115G>A	p.V39M	Null	−0.69	Decrease	−0.63162291	Decrease	0.620853
	Human	c.358C>T	p.P120S	Decrease	−1.83	Decrease	−0.33095521	Decrease	0.887135

* *NPC1*:c.3224G>A (p.R1075H) mutation was identified in a Siamese cat with Niemann–Pick disease type C (NPC) in this study. ** *NPC2*:c.376G>A (p.V126M) mutation was identified in Siamese and Japanese domestic cats with NPC in this study. The other mutations have also been reported to be pathogenic or NPC in cats, cattle, and humans. ddG: delta delta G, Null: no change in stability, Increase: increased stability, Decrease: decreased stability, and Conf. score: confidence score.

## Data Availability

Not applicable.

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
