# Peer review of "Novel Mutation in the Feline NPC2 Gene in Cats with Niemann–Pick Disease"

_animals, 2023, doi:10.3390/ani13111744_

Round 1
Reviewer 1 Report
Well done. I think they could better support the splice variant as causal.
This study is very well done, thorough, easy to read and concise. The only thing missing is population data, which I offer.
The 99 Lives Cat Genome Consortium data is free for use to the scientific community. To use the data below, just put in the acknowledgements – “We appreciate the variant frequency data provided by the 99 Lives Consortium”.
In general, the NPC1 and NPC2 variants were absent in the 99 Lives WGS data from 362 domestic cats and WES data from 62 cats, suggesting these variants are rare and support causality. Only one variant was identified in the 99 Lives dataset, the NPC1 variant c.54G>T (p.A18A) was common and found to be homozygous 31 cats and heterozygous 130 cats. However, additional Asian cats would further support the 99 Lives consortium dataset.
(However, please note - my annotation uses NPC2 transcript XM_003987833.5 and NPC1 XM_019814307.2 – be certain your positions are the same in these transcripts)
XM_019814307.3 (NPC1) and XM_003987833.6 (NPC2)
In NPC1:
c.54G>T (p.A18A) – common 31 homozygous and 130 heterozygous
c.1815T>C (p.T605T) - absent
c.2214C>A (p.S605S) - absent
(c.2043C>T (p.P681P) – absent
(c.3224G>A (p.R1075H) - absent
In NPC2:
c.376G>A, p.V126M) – absent
c.364-4C>T (g.121865226C>T) - absent
For the silent mutation (c.588A>C (p.S66S)) – there is an error here - 66 x 3 = 198 (588/3 = 196)
Add splice to the simple summary
Need a table of the primers and PCR conditions sufficient to replicate study.
L128 above mentioned – add space
L130 add “the” – from the two affected cats
Well done – showing results of other pathogenic variants!
Why c.3224G>A p.R1075H not in Table 2? Add to be complete – more support.
In regards to the NPC2:c.364-4C>T (g.121865226C>T:
See this website: https://www.ncbi.nlm.nih.gov/clinvar
Consider these 2 variants below and their disease course in humans – maybe some more support for your variant?
NM_006432.5(NPC2):c.364-2A>G
NM_006432.5(NPC2):c.364-3C>T
Maybe call this a variant of “uncertain significance” if appropriate.
Please comment to the amplification of FFPE tissue? An example of the difficulties you faced would be most useful. Any modifications to PCR to get the amplifications to work? How much tissue did you try to use to get the DNA? More details to these methods would certainly help others! How many sequences did you produce to get consensus? These details are important.
Please note – you do not have to use the 99 Lives data, that is not a condition of acceptance. However, population data would be very supportive of the claims!
Reviewer 2 Report
This manuscript is describing identification of novel candidate mutations for Niemann Pick disease in two cats diagnosed pathologically. The authors found two different mutations in two cats with NP disease by using genomic DNA prepared from formalin-fix paraffin embedded (FFPE) sections. Prediction of protein function was performed all mutation found in this study. The study design, methodology and interpretation of results are appropriate. No other experiments are needed for publication of this manuscript.
The result from the Siamese cat is strait forward. For the JD cat, a few examinations such as RT-PCR which could be performed with total RNA from FFPE section are needed to clarify whether another mutation is present in an intron or c.364-4C>T may induce an aberrant splicing event despite low possibility. Once later rules out, another mutation is most likely. Even so, there is no doubt that c.367G>A leads to protein dysfunction. Pregnant female cats living outdoor may intake some mutagenic substances during pregnant, which could introduce mutations to the genomic DNA of fetuses. If stray pregnant cats with heterozygous c.367G>A gained unknown mutations in NPC2, compound heterozygous cats for NPC2 would be produced. I think that this is an important issue for veterinary clinicians. This would not be a situation unique to Japan. Gene frequency of c.367G>A needs to be examined in the future. Therefore, I would appreciate to add discussion why the Siamese cat is homozygote but the JD cat is compound heterozygote.
